# Multi-generational koala pedigree analysis reveals rapid changes in heritable provirus load associated with life history traits

Guilherme B. Neumann [1], Rachael Tarlinton [2], Paula Korkuć [1], Patricia M. Gaffney[3,12], Karine A. Viaud-Martinez [4], Sylwia Urbaniak[4], Kan Nobuta[4], Anisha Dayaram [1,5], Baptiste Mulot [6], Kerstin Ternes[7], David E. Alquezar-Planas [1,8], Alfred L. Roca[9], Patric Jern [10], Cora L. Singleton [3] ✉ & Alex D. Greenwood [1,11] ✉

Retroviruses that colonize the host germline can be passed on as inherited genetic variants. The koala (*Phascolarctos cinereus*) is currently experiencing germline colonization by two retroviruses, the koala retrovirus (KoRV) and phaCin-β. We analyze the integration site segregation and diversity of endogenous KoRV, phaCin-β, and the related phaCin-β-like in 111 pedigreed koalas from the San Diego Zoo Wildlife Alliance and seven European Zoos. The use of multigenerational pedigrees and the inclusion of health information for each individual koala reveal elimination of retroviruses from proto-oncogenes and the generation and spread of new germline integrations. Seven-hundred-and-fourteen integrations do not persist in the living population. For the 55 triads examined, 21 unique integrations identified in individual koalas are absent in their parents. Retroviral integrations associated with leukemia, fertility, and longevity are used to estimate genetic risk scores and develop a longevity breeding index to minimize neoplasia risk in the captive koala population.

Retroviruses are single stranded RNA viruses that integrate a reverse transcribed DNA provirus copy of their genome into host nuclear DNA during viral replication. If this occurs in the germline, the resulting provirus can be transmitted in a Mendelian fashion. Such germline proviral sequences are called endogenous retroviruses (ERVs). The koala (*Phascolarctos cinereus*) has experienced multiple recent and ongoing germline colonizations by retroviruses, including the Koala Retrovirus (KoRV)[1] and phaCin-β[2], which may still co-exist with exogenous (infectious) retroviral lineages whereas other endogenized retroviruses, such as phaCin-β-like[2] and the Phascolarctos endogenous

retroelement (PhER)[3] do not. KoRV is a gammaretrovirus most closely related to the gibbon ape leukemia viruses (GALVs). It was first characterized from koala lymphoma tissues and in mitogen-stimulated peripheral blood mononuclear cell cultures from 163 koalas[1]. KoRV was subsequently determined to be a vertically transmitted ERV that had only recently colonized the koala, as populations that were KoRV free were identified[4]. Based on the DNA mutation rate per base pair per year of humans ($2 \times 10^{-9}$) and mice ($4.5 \times 10^{-9}$), KoRV was estimated to have begun colonizing the koala germline between 22,200 and 49,900 years ago[5]. The phaCin-β and phaCin-β-like ERVs were estimated to

¹Department of Wildlife Diseases, Leibniz Institute for Zoo and Wildlife Research, Berlin, Germany. ²School of Veterinary Medicine and Science, University of Nottingham, Sutton Bonington Campus, Loughbrough, UK. ³San Diego Zoo Wildlife Alliance, San Diego, CA, USA. ⁴Illumina Laboratory Services, Illumina Inc., San Diego, CA, USA. ⁵Institute for Neurophysiology, Charité-Universitätsmedizin Berlin, Berlin, Germany. ⁶Zooparc de Beauval & Beauval Nature, Saint-Aignan, France. ⁷Zoo Duisburg GmbH, Duisburg, Germany. ⁸Australian Museum Research Institute, Australian Museum, Sydney, NSW, Australia. ⁹Department of Animal Sciences, University of Illinois at Urbana–Champaign, Urbana, IL, USA. ¹⁰Department of Medical Biochemistry and Microbiology, Uppsala University, Uppsala, Sweden. ¹¹Department of Veterinary Medicine, Freie Universität Berlin, Berlin, Germany. ¹²Deceased: Patricia M. Gaffney. ✉e-mail: CSingleton@sdzwa.org; greenwood@izw-berlin.de

have begun colonizing the koala germline between 0.7 and 1.5 Mya, and between 2.4 and 5.5 Mya, respectively[2].

The number of KoRV and phaCin-β loci per individual varies among populations, in contrast with most other vertebrate ERVs where the number remains constant. The variation in the number of integrations is consistent with both ERVs being in the early stages of endogenization. Koalas in northern Australia generally carry more endogenous KoRVs (enKoRVs), with numbers ranging from 54 to 116 per individual. In the south, koalas carry fewer enKoRVs, ranging from 8 to 23 per individual[6]. The absence of enKoRVs was previously reported on Kangaroo Island, south of South Australia[4]. A less pronounced north-south trend was reported for phaCin-β, ranging from 48 to 90 phaCin-β integrations in the north to range of 67–81 phaCin-β in the south[6]. The number of ERVs per individual for phaCin-β-like ranged from 46 to 64 polymorphic phaCin-β-like integrations in the north, and from 53 to 67 phaCin-β-like integrations in the south[6]. The more widespread stability of copy number and location of phaCin-β-like is consistent with germline colonization predating both phaCin-β and enKoRV.

There are health consequences associated with proviral integrations. For the koala, it has been reported up to 60% die from KoRV associated leukemia and lymphoma in captive populations[1,7]. Provirus integrations into enhancer or promoter sequences can activate adjacent gene expression. Proviruses can also influence mRNA 3′-end processing when downstream of a gene or can inactivate genes by insertional disruption[8,9]. Tumor tissues accumulate somatic KoRV integrations which are significantly overrepresented in proximity to cellular proto-oncogenes[10].

Most studies of KoRV and other koala retroelement endogenization have been conducted at the population level without pedigree information. Using pedigrees, it is possible to determine the frequency of novel germline integrations, how and which potentially deleterious ERVs are lost, and examine whether positive selection for protective ERVs is detectable. Pedigree data allow for the determination of occurrence and dosage of ERV alleles in families. For example, new ERV integrations can be unique to offspring but absent in their parents. Additionally, metadata on life and health histories of each individual can associate information on relatedness with impacts of specific ERVs on the fitness and survival of individuals.

In this study, we characterize the diversity of enKoRV, phaCin-β, and phaCin-β-like ERVs in koalas from the San Diego Zoo Wildlife Alliance (SDZWA), utilizing the complete pedigree information to investigate the evolutionary dynamics of ERVs for up to four koala generations. Comparisons with wild koalas indicate that the SDZWA koalas are genetically more closely related to northern Australian koalas than to koalas from South Australia. In addition, there is evident genetic relatedness between SDZWA koalas and koalas from European zoos (EUZ), as expected due to translocations between zoos in North America and Europe. We identify instances of ERVs that do not persist across generations, and the appearance of new KoRV and phaCin-β proviruses in the germline. Additionally, we identify candidate markers, including ERVs and single nucleotide polymorphisms (SNPs), associated with traits, such as leukemia, fertility, and longevity, making them important markers for consideration in captive breeding programs. This analysis forms the basis of determining genetic risk scores for the captive SDZWA koala population.

## Results
### Koala pedigrees from the SDZWA and EUZ
Koalas were first brought to North America between 1976 and 1981, comprising 14 founders from the Lone Pine Koala Sanctuary. After 1990, occasional imports occurred, including nine koalas from Japan, Currumbin, and Coomera. SDZWA currently keeps 30 koalas, the largest captive population outside Australia. Pedigree information for all animals comprises information about parentage data, sex, date of birth, birth location, date of death, and location and cause of death (if deceased). In addition to blood samples from all living animals, cryopreserved tissues of koalas collected over 40 years were available for DNA extraction for a total of 91 koala individuals, including three individuals with different tissue replicates. This sample collection represents four generations of koalas, dating back to animals born in 1978 (Fig. 1a), including one of the original founders (Studbook number (SB) 106). It comprised 46 triads, including 19 F2 and five F3 generations (Supplementary Data 1).

Twenty koalas from seven EUZ were also used in the investigation of enKoRV only (Supplementary Data 2). These comprised nine triads, including two F2 generations. Individuals from EUZ are related to individuals from SDZWA, with two individuals having parents from

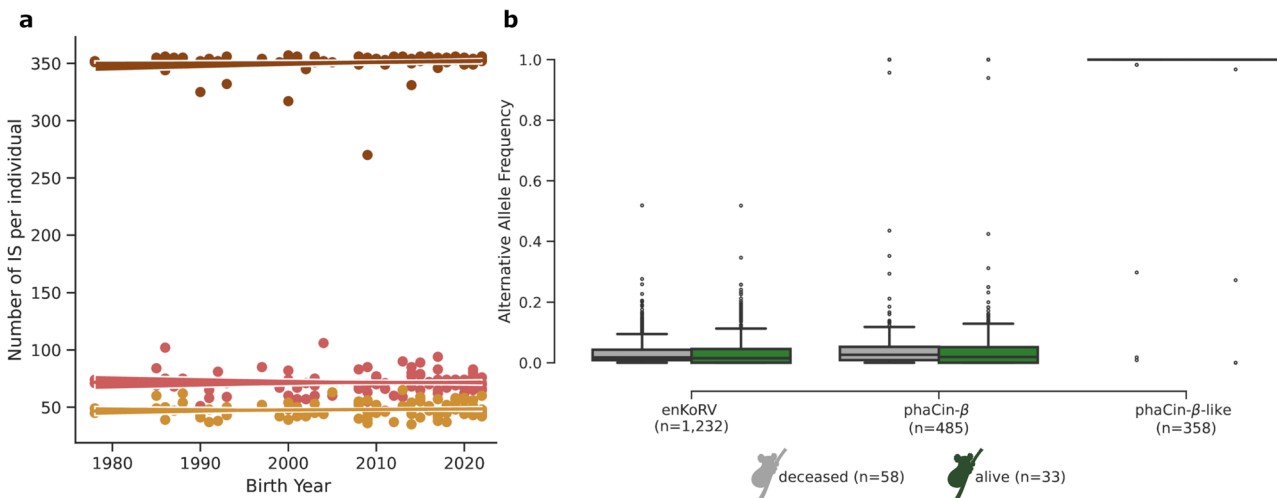

**Fig. 1 | Distribution of detected enKoRV, phaCin-β, and phaCin-β-like integrations. a** Number of ERVs per individual plotted against individual birth years. A linear regression line was added to illustrate the trend in the data and shaded band is the 95% confidence interval around the fit. **b** Boxplot distribution of alternative allele frequencies for deceased (gray) and living (green) individuals. Boxplot shows the median, the 25–75th percentiles in the box, and whiskers to 1.5× IQR

(interquartile range). Outliers outside the whiskers are represented as dots. The y-axis indicates the frequency of the alternative allele (integration) of the different endogenous retroviruses, where 0 indicates the retrovirus integration is absent in the group of living or deceased animals and 1 indicates fixation of the integration in the group of living or deceased animals. Source data are provided as a Source Data file.

SDZWA (SB 296 and 367), and individuals having grandparents from SDZWA (SB 201, 206, 244, 264, 399, 403, 436, and 537).

## enKoRV and phaCin-β polymorphic integrations

A total of 2075 ERV integrations (1232 enKoRVs, 485 phaCin-β, and 358 phaCin-β-like) were characterized across 91 koala individuals from the SDZWA population. On average, each koala had 471 ERV integrations, ranging from 379 to 510. On average, there were 72 enKoRV, 48 phaCin-β, and 351 phaCin-β-like per koala. From 1978 to 2022, the number of ERV integrations per individual was stable (Fig. 1a).

Most of the phaCin-β-like, which are the oldest of these ERVs, were fixed in the population except two integrations that continue to segregate in the living population (Fig. 1b). Alternative Allele Frequency (AAF) was calculated considering the pre-integration site as reference allele and the ERV as the alternative allele. One of the segregating phaCin-β-like was intergenic and observed in 52 individuals (AAF = 0.27), including the founder individual, while the other was close to fixation (AAF = 0.97), located in an intron of the oncogene *SLC12A7* (solute carrier family 12 member 7). *SLC12A7* encodes a kinase binding protein, with overexpression associated with aggressive adrenocortical carcinoma in humans[11]. Both enKoRV and phaCin-β integrations displayed significantly lower AAFs than phaCin-β-like, except for a few phaCin-β, which were observed at higher frequencies that were mostly homozygous (Supplementary Figs. 1 and 2).

## Germline mutation rate and retroviral germline colonization timeline

SNPs present in koala offspring (joeys) and not present in their parents in 46 triads were used to estimate the koala mutation rate. After strict filtering of new SNPs, to avoid false-positives as described in the Methods, a mutation rate per base pair per generation of $1.03 \times 10^{-8}$ was estimated. This is slightly lower than the estimated rate for humans ($1.13 \times 10^{-8}$) and cattle ($1.17 \times 10^{-8}$). The mutation rate per base pair per year was estimated to be $1.6 \times 10^{-9}$, lower than the annual estimated rate for humans ($2.0 \times 10^{-9}$) and mice ($4.5 \times 10^{-9}$), used previously to estimate the initial germline colonization of KoRV of not more than 50,000 years ago, phaCin-β 1.5 million years ago (Mya) and phaCin-β-like and 5.5 Mya.

Using the lower koala mutation rate, we revised KoRV germline colonization start to not more than 312,191 years ago, concomitant with the emergence of modern koalas[12]. Germline colonization estimates for phaCin-β and phaCin-β-like with our revised koala mutation rate resulted in colonization times of 1.9 and 6.9 Mya, respectively. Nevertheless, younger estimated dates cannot be discarded due to the possibility of gene conversion or other recombination events[13].

## Origin of SDZWA and EUZ ERVs

In total, 75 wild koalas from the Koala Genome Survey (Supplementary Data 3 and Supplementary Fig. 3) were compared with captive koalas from the SDZWA. Twenty-five individuals derived from each state, Queensland (QLD), New South Wales (NSW) and Victoria (VIC), carrying 2616 enKoRVs, 1446 phaCin-β, and 359 phaCin-β-like. Of these 4421 ERVs, 383 were shared between VIC and SDZWA, 582 between NSW and SDZWA, and 837 between QLD and SDZWA (Fig. 2a). The number of shared ERVs was inflated by the 356 phaCin-β-like common to all populations (Supplementary Fig. 5). Fewer shared integrations were observed for enKoRV (Fig. 2b) and phaCin-β (Supplementary Fig. 6). Notably, the low number of enKoRVs in VIC (*n* = 110) is consistent with other studies demonstrating low KoRV prevalence in VIC and South Australia (Fig. 2b)[6]. Only two enKoRVs from VIC were shared with SDZWA. However, in comparison to SDZWA, the AAFs of phaCin-β in VIC were higher (Supplementary Fig. 7).

The number of ERVs per individual differed among populations. Koalas from QLD had on average 72 enKoRVs, 70 phaCin-β, and 326 phaCin-β-like. Similarly, NSW koalas had on average 73 enKoRVs, 82 phaCin-β, and 336 phaCin-β-like. The average numbers were similar to koalas from SDZWA, except for phaCin-β, which was more frequent in wild koalas. A different pattern was observed in VIC with an average of 8 enKoRVs, 71 phaCin-β, and 337 phaCin-β-like per koala. This result corroborates previous findings on the low prevalence of enKoRV in VIC and South Australia[6]. Two koalas from Bawbaw and Strathbogies (VIC) had only one enKoRV each. Nevertheless, differences in sequence coverage may have influenced detection of ERVs in these individuals. A significant Pearson correlation between number of phaCin-β-like per individual and coverage was observed for NSW ($r = 0.72$, $p$-value = $4.4 \times 10^{-5}$) and VIC ($r = 0.69$, $p$-value = $1.2 \times 10^{-4}$), and of enKoRV per individual and coverage for VIC ($r = 0.59$, $p$-value = $1.9 \times 10^{-3}$).

The 130 shared enKoRV between QLD and SDZWA reflects the founding history of the SDZWA koala population, with 14 individuals coming from the Lone Pine Koala Sanctuary in Brisbane (QLD). Imports of individuals from NSW have occurred, accounting for the shared ERVs between SDZWA and NSW koala populations.

While koalas from SDZWA cluster separately in the PCA (Fig. 2c), three individuals show close relationship with QLD and NSW koalas, including two individuals transferred to SDZWA (SB 267 and 438) and one offspring (SB 153) of a founder koala (SB 106). Individuals from QLD and NSW show overlap of ERV integrations, likely explained by gene flow between the regions. Individuals from VIC form a distinct cluster, emphasizing the isolation of koalas from this region. However, one individual from southern NSW, Narrandera, clustered with VIC koalas rather than NSW koalas, explained by the mixed ancestry of koalas in this region reintroduced from QLD and VIC in 1970s[14].

Twenty koalas from different EUZ were targeted for KoRV enrichment[15], resulting in the discovery of 2310 proviral integrations that we characterized as 553 enKoRVs and 1757 somatic KoRV integrations (see Methods). On average, each individual had 97 enKoRVs, ranging from 42 to 141, and 88 somatic KoRVs, ranging from 3 to 241. While most enKoRVs in koalas from EUZ were unique, 20% (244) were shared with SDZWA. Similarly, enKoRVs were shared between koalas from QLD and EUZ and between koalas from NSW and EUZ.

The enKoRV with the highest frequency in all populations (AAF = 0.43) was the intronic enKoRV located in the oncogene *SLC29A1* (solute carrier family 29 member 1 - Augustine blood group). Another enKoRV was located upstream of *SLC29A1* (AAF = 0.07). Further enKoRVs high AAFs (AAF > 0.10) were located inside intronic regions of the genes *PCCA* (propionyl-CoA carboxylase subunit alpha), encoding the alpha subunit of the heterodimeric mitochondrial enzyme Propionyl-CoA carboxylase, and *PTPRT* (protein tyrosine phosphatase receptor type T), encoding a tumor suppressor protein. For phaCin-β, regions with high AFFs were located in genes, such as *TMEM169* (transmembrane protein 169), *KCTD21* (potassium channel tetramerization domain containing 21), *KCNK2* (potassium two pore domain channel subfamily K member 2), *ORC4* (origin recognition complex subunit 4), and *SNRPC* (small nuclear ribonucleoprotein polypeptide C), fixed in both captive and wild populations (AAF = 1.00).

Non-singleton enKoRVs (1595 enKoRVs shared by at least two individuals) were analyzed using PCA including EUZ (Fig. 2d). With EUZ data, only the detection or absence of enKoRV was used, since the sequencing data for EUZ was targeted for KoRV. Captive and wild koalas formed distinct clusters, with a clear relationship between koalas from SDZWA and EUZ.

Although most ERVs identified in our study were intergenic (4479), we identified 1273 intronic ERVs. Further, 26 enKoRV and one phaCin-β were observed in coding sequence (CDS) regions. Of the 26 enKoRVs integrations in CDS regions, one was classified as a start loss of a novel gene (gene ID g21247) located on contig JAOEJA010000073.1 at 15,262,207 bp. However, this was observed in only one of the SDZWA koalas (SB 127, born in 1985) and was not inherited by its offspring. The remaining 25 ERVs in CDS regions were classified as feature

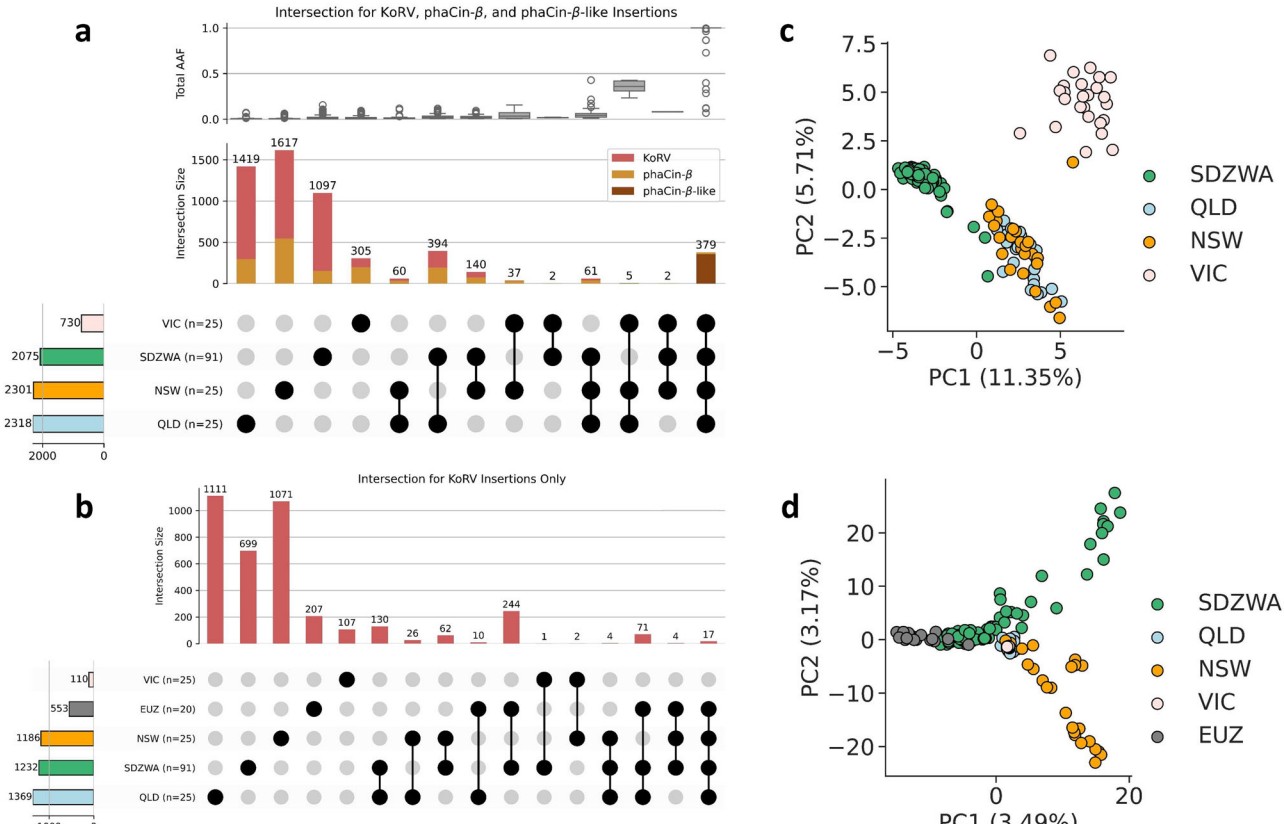

**Fig. 2 | ERV relationship between wild and captive koalas. a** Upset plot for the intersection between wild populations from Queensland (QLD), New South Wales (NSW) and Victoria (VIC), and San Diego Zoo (SDZWA) populations, for enKoRV, phaCin-β, and phaCin-β-like. Numbers in parenthesis indicate number of animals per population used in the analysis. Circles filled in black represent the intersections, with respective bars indicating the number of enKoRV (red), phaCin-β (yellow), and phaCin-β-like (brown) shared between the marked populations. The left bar shows the total number of phaCin-β observed for each population, while colors represent the different populations. The boxplot above shows the total Alternative Allele Frequency (AAF) calculated for all populations together, and is shown for each intersection group. Boxplot shows the median, the 25–75th percentiles in the box, and whiskers to 1.5× IQR (interquartile range). Outliers outside the whiskers are represented as dots. **b** Upset plot for the intersection between wild, SDZWA, and European zoos (EUZ) populations, for enKoRVs only. **c** PCA considering the genotypes of all ERVs with an AAF ≥ 0.05 of SDZWA and wild populations. **d** Standardized PCA considering the presence or absence of non-singleton enKoRVs of wild, SDZWA, and EUZ populations. Source data are provided as a Source Data file.

elongations, and only seven were observed in more than one individual, with the highest AAF = 0.05. One ERV was located in the *MCM9* gene, a gene encoding a mini-chromosome maintenance (MCM) protein that is indispensable for the initiation of eukaryotic genome replication. Mutations in this gene have been associated with ovarian failure, short stature, and chromosomal instability[16].

### ERV integrations over generations
The AAFs of deceased and living animals were different for some enKoRV and phaCin-β. In total, 714 ERVs were absent in the living population which suggests an elimination rate of $9.58 \times 10^{-10}$ ERVs per base pair per generation for enKoRV and $2.51 \times 10^{-10}$ ERVs per base pair per generation for phaCin-β. For example, the previously described enKoRV in the exon of the oncogene *BCL2L1* (BCL2 like 1)[10] was detected in four deceased animals, all heterozygous, but in none of the living animals. The four animals in three distinct triads died of leukemia. The integration was not transmitted further and disappeared from the pedigree (Fig. 3).

Most ERVs that did not persist in the current captive population were integrations restricted to single individuals that were not transmitted to offspring. However, 37 ERVs (25 enKoRVs and 12 phaCin-β) occurred with a higher frequency (AAF$_{deceased}$ > 0.05) but are also now absent from the current captive population (Supplementary Data 4). For example, an enKoRV located 8 kbp downstream of the *ALDH1A3* (aldehyde dehydrogenase 1 family member A3) gene was present in

seven individuals. Despite the fact that those koalas reached ages between 8 and 15 years, only two of them had four and five offspring, respectively, of which three were sequenced (Fig. 4). Of those three sequenced offspring, only one individual inherited the enKoRV, but did not transmit it to the next generation since it died of peritonitis, colitis, and metabolic imbalance five months after birth. Mutations in this gene are associated with fetal malformations in humans[17], including anophthalmia/microphthalmia and facial dysmorphic features.

### New ERV integrations
Across the 46 triads in the SDZWA, we detected 16 joey-specific ERV integrations, 15 enKoRVs and one phaCin-β, at an estimated 0.33 KoRV and 0.02 phaCin-β germline integrations per generation. The number of new ERVs varied from one (in three cases, SB 472, 560, and 748), two (in four cases, SB 551, 657, 665, and 674), to five (in one case, SB 659) per joey. Five of 16 new ERVs (20% probability) were detected in the subsequent generation. These included one new phaCin-β (SB 472), two new enKoRVs in the same joey (SB 674)—one of which was located in an intron of the *DDAH1* (dimethylarginine dimethylaminohydrolase 1) gene—and two new enKoRVs detected in another joey (SB 659). The expected number of transmitted new ERVs was seven, given a 45% inheritance rate when considering a 10% probability of missing heterozygous ERVs based on a binomial model given a coverage ≥ ten sequencing reads. The observed transmission

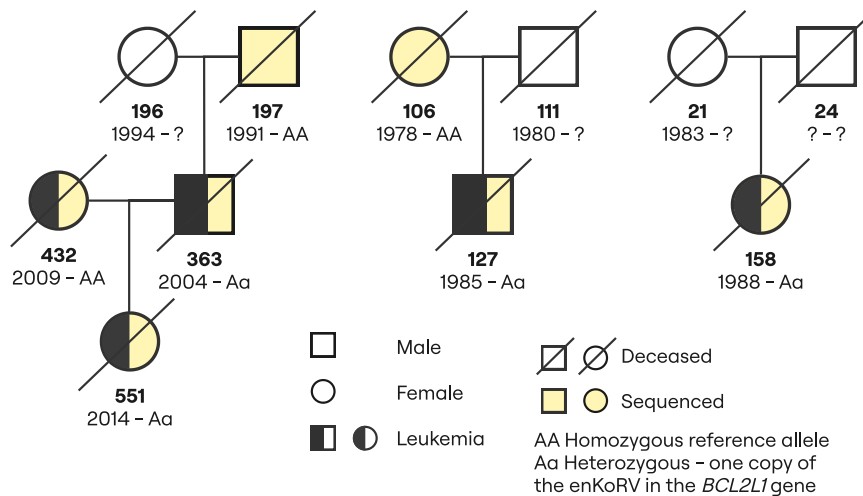

**Fig. 3 | Pedigree of carrier koalas for the enKoRV located on the exon of the oncogene *BCL2L1*.** Below each animal is the studbook number, followed by birth year and the genotype code. Death or euthanasia due to leukemia (where known) is indicated by a pattern.

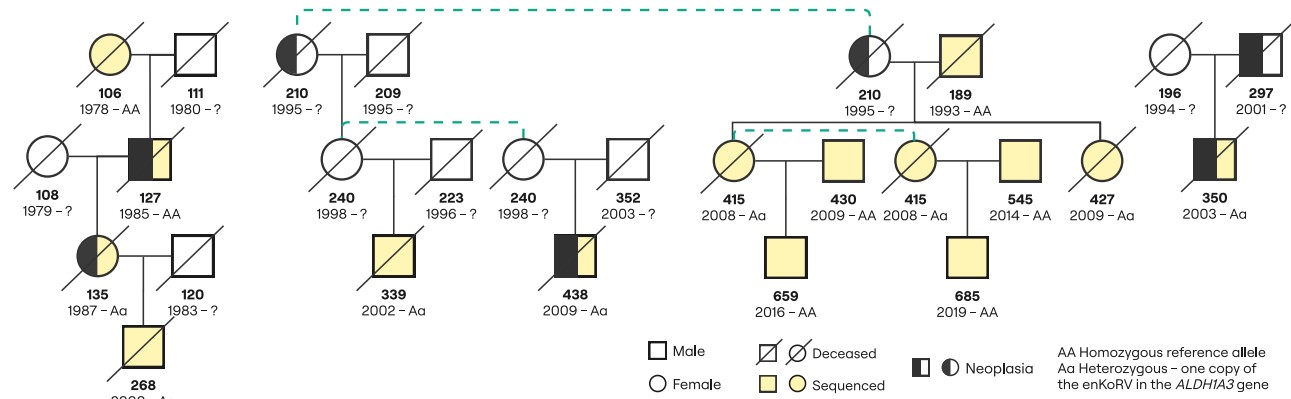

**Fig. 4 | Pedigree of carrier koalas for the enKoRV located 8 kbp downstream the gene *ALDH1A3*.** Below each form is the studbook number, followed by birth year and the genotype code. Cause of death or euthanasia (where know) is indicated by a pattern. The dashed lines indicate cases where the same female had different mate partners.

rate of five is lower than but still consistent with the accepted Mendelian inheritance rate.

Across the nine EUZ koala triads, we detected five joey-specific enKoRV germline integrations. The number of new ERVs varied from one (three cases, SB 395, 429, and 475) to two (one case, SB 369) integrations per joey, representing 0.56 germline integrations per generation. Only one new enKoRV was transmitted to the F2 generation (from SB 395 to SB 475). However, only two joeys had data available for the F2 generation and it is thus not possible to determine if the transmission is consistent with Mendelian inheritance. Two of the enKoRVs were located in the flanking regions of two genes, *PTPN21* (protein tyrosine phosphatase non-receptor type 21), as part of the protein tyrosine phosphatases family, known to regulate a variety of cellular processes including cell growth, differentiation, mitotic cycle, and oncogenic transformation[18], and *MGAT5* (alpha-1,6-mannosylglycoprotein 6-beta-N-acetylglucosaminyltransferase) a mediator of cell migration in tumor development[19,20].

**enKoRVs can dysregulate gene expression**

Three genes containing commonly shared intronic enKoRV in the captive populations were differentially expressed in comparison to koalas without these enKoRVs. RNA-seq data was available for six of the EUZ koalas and five publicly available KoRV-free wild koalas[10,21]. Four of the EUZ individuals containing the most common intronic

enKoRV in the *SLC29A1* gene showed significantly higher *SLC29A1* expression than individuals without the enKoRV (Fig. 5a). A full length KoRV-A was integrated in intron 9 of *SLC29A1*.

Two other genes upregulated due to the presence of an intronic enKoRV were *LONP2* (Lon Peptidase 2) and *STAP1* (Signal Transducing Adapter Family Member 1). *LONP2* encodes for a peroxisome enzyme responsible for selectively degrading oxidatively damaged proteins, and is found to be upregulated in various cancers, in particular cervical cancer[22]. *STAP1* is a gene that encodes for a protein involved in intracellular signaling by facilitating interactions between other proteins within signal transduction pathways. *STAP1* has been linked to B-cell receptor signaling, which is crucial for the host immune response[23]. Moreover, *STAP1* expression in glioma-associated microglia was positively correlated with the degree of malignancy and poor prognosis of glioma[24].

In SDZWA, 18% of the koalas carried the enKoRV in the *LONP2* gene, while 60% of the EUZ koalas carried the same enKoRV. In the case of the enKoRV in the *STAP1* gene, a prevalence of 11% in SDZWA and 25% in EUZ was observed. The enKoRVs were absent from wild populations examined.

**enKoRVs and phaCin-β are associated with increased cancer rates in koalas**

Based on cause of death information for each koala, cancer rates were compared among deceased animals (n = 58). In total, 27 individuals

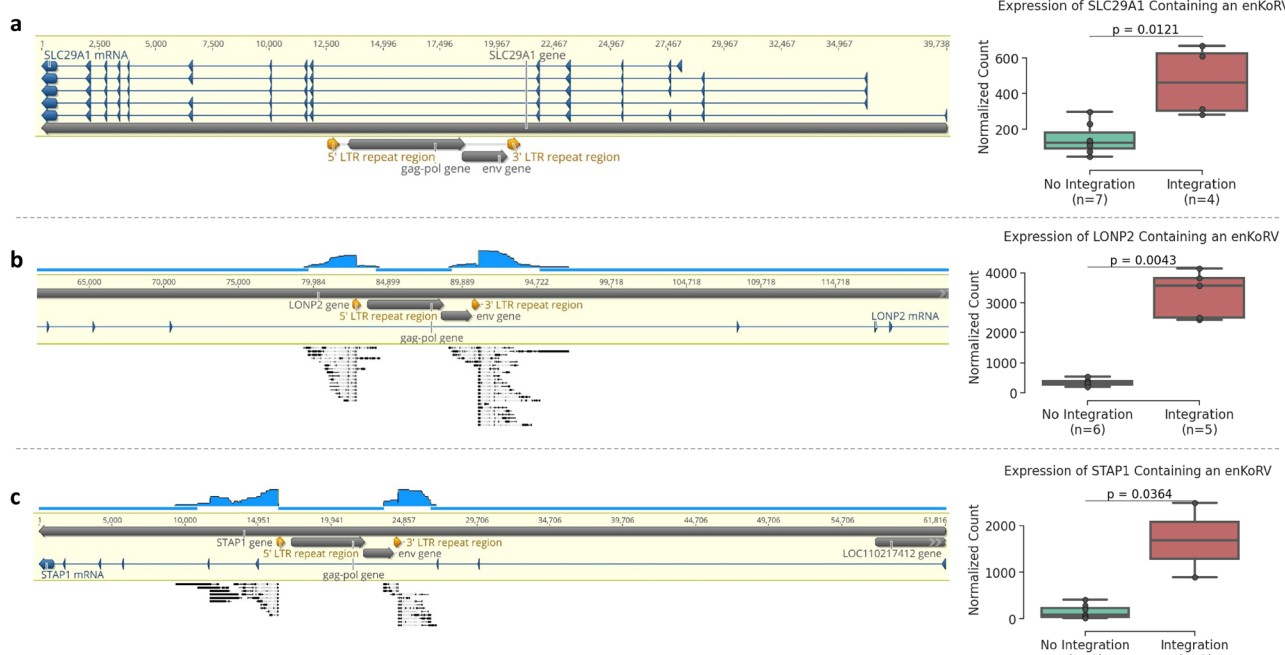

**Fig. 5 | Three genes upregulated due to an intronic enKoRV. a** Sequence of the koala reference genome, Bilbo, for the *SLC29A1* gene containing an enKoRV in the intron region. **b** Sequence of the *LONP2* gene containing an enKoRV-A integration in an intron region, mapped to the contigs of PacBio long-reads assembled from an EUZ koala. **c** Sequence of the *STAP1* gene containing an enKoRV-A integration in an intron region, mapped to the contigs of PacBio long-reads assembled from an EUZ koala. On the right side of each alignment sequence, a boxplot shows the gene count normalized by sequencing depth in animals containing the enKoRV versus the normalized gene count in animals without the enKoRV. Boxplots show the median, the 25–75th percentiles in the box, and whiskers to 1.5× IQR (interquartile range). A significant *p*-value of a two-side Mann-Whitney test comparing the two groups is shown above the plots. Source data are provided as a Source Data file.

died of neoplasia, including diagnosis of lymphoma ($n = 13$) and/or leukemia ($n = 11$), sarcoma ($n = 6$), osteochondroma ($n = 4$), and adenocarcinoma ($n = 1$). Overall, 34 enKoRVs showed an increased prevalence of more than 10% in cancer cases compared to other causes of death (Supplementary Data 5). Those 34 enKoRVs were located inside or in the flanking regions ( ± 10 kb) of 15 genes, including known proto-oncogenes, such as *BCL2L1, SLC29A1*, and *LZTS1* (leucine zipper tumor suppressor 1). In the case of phaCin-β, 14 showed a higher prevalence in neoplasia cases flanking three genes: *DOCK8* (dedicator of cytokinesis 8)*, FAM216B* (family with sequence similarity 216 member B), and *Vmn2r65* (vomeronasal 2, receptor 65).

To estimate the direction and magnitude of ERV effects on neoplasia, a logistic regression was performed in a case-control model testing leukemia cases versus other death causes (associations with $p \leq 0.05$, Table 1). Seven alleles had an odds ratio (OR) > 1 indicating a higher probability of developing leukemia. Interestingly, individuals without the enKoRV located in the intron of *SLC29A1* gene, which has the highest enKoRV AAF in the koala population, showed 4.9-fold higher chances of developing leukemia than individuals carrying this enKoRV. The enKoRV in the exon region of *BCL2L1*, which is no longer present in the current captive population, showed a 21-fold higher chance of developing leukemia. The enKoRV in the intron region of the proto-oncogene *LZTS1* had an OR of 9.3. Two phaCin-β integrations were potentially protective (OR < 1), one located downstream of *GIN1* (gypsy retrotransposon integrase 1) gene.

### ERVs and SNPs are associated with fertility and longevity

Reproductive success (presence or absence of offspring) was tested using logistic regression. For longevity estimate, age at death was tested using a linear regression. In total, 24 deceased koalas never had offspring, while 34 had at least one. No SNP was significantly associated with the reproductive success trait, but 18 ERVs suggested association (associations with p ≤ 0.05, Table 2). Two ERVs were associated with higher reproductive success, one enKoRV in the proximity of the genes *RAB13* (RAB13, member RAS oncogene family)*, NUP210* (nucleoporin 210), and *RPS27* (ribosomal protein S27), and one intergenic phaCin-β. The remaining ERVs showed decreased likelihood of reproductive success. The eliminated enKoRV downstream of *ALDH1A3* also showed an OR < 1, meaning lower chance of reproductive success. Another phaCin-β in the *Vmn2r65* gene, close to the phaCin-β with 2.3-fold OR for leukemia in the intron of the same gene, also decreased the chances of reproductive success. However, it was less frequent in the population (AAF = 0.05) than the phaCin-β associated with leukemia (AAF = 0.44). Although not significant, the intronic enKoRV in the *LONP2* (Lon Peptidase 2) gene, which caused an upregulation of the *LONP2* gene, showed a *p*-value of 0.06 for having offspring with an OR = 0.21 (Supplementary Data 6). Eight SNPs, 48 enKoRVs, and 13 phaCin-β were suggestively associated with age at death (Supplementary Fig. 9). Identified SNPs were in the introns of *SPATA21* (spermatogenesis associated 21), *NECAP2* (NECAP endocytosis associated 2), and *CROCC* (ciliary rootlet coiled-coil, rootletin) genes. Though not reaching statistical significance, the ERVs were found in the flanking region of genes, such as *FANCI* (FA complementation group I), associated with fanconi anemia[25], and *FAP* (fibroblast activation protein alpha) and *CX3CL1* (C-X3-C motif chemokine ligand 1), candidates for polyarteritis nodosa, a disease causing inflammation of the arteries[26]. *CX3CL1* has also been linked to inflammatory diseases and cancer[27].

### Genetic risk score estimation

For two binary traits analyzed, leukemia and reproduction success, a genetic risk score was calculated based on the markers detected with a suggestive association. An average prediction accuracy of 0.64 and 0.80 was detected for reproduction success and leukemia, respectively, applying 100 bootstrapping repeats. The cutoffs (Fig. 6) were determined based on the highest accuracies generated.

**Table 1 | Associations of ERVs with leukemia cases**

| ERV | Contig | Position | AAF | Effect allele | OR | Up or down | p-value | Gene | Region |
|---|---|---|---|---|---|---|---|---|---|
| enKoRV | JAOEJA010000643.1 | 3,697,700 | 0.52 | Ref | 4.9 | Up | 0.01 | *SLC29A1* | intron |
| enKoRV | JAOEJA010000643.1 | 3,684,723 | 0.08 | Alt | 5.0 | Up | 0.04 | *MYMX, SLC29A1* | downstream |
| phaCin-β | JAOEJA010000944.1 | 5,007,950 | 0.44 | Alt | 2.3 | Up | 0.04 | *Vmn2r65* | intron |
| phaCin-β | JAOEJA010000514.1 | 105,854,631 | 0.14 | Alt | 0.1 | Down | 0.04 | - | intergenic |
| enKoRV | JAOEJA010000639.1 | 20,432,126 | 0.03 | Alt | 21.0 | Up | 0.05 | *BCL2L1* | exon |
| phaCin-β | JAOEJA010000514.1 | 147,057,845 | 0.13 | Alt | 0.1 | Down | 0.05 | *GIN1* | downstream |
| enKoRV | JAOEJA010000656.1 | 102,559,343 | 0.04 | Alt | 9.6 | Up | 0.05 | - | intergenic |
| phaCin-β | JAOEJA010000656.1 | 97,917,242 | 0.05 | Alt | 9.3 | Up | 0.05 | - | intergenic |
| enKoRV | JAOEJA010000379.1 | 3,475,542 | 0.04 | Alt | 9.3 | Up | 0.05 | *LZTS1* | downstream |

*ERV* Endogenous retrovirus, *AAF* Alternative allele frequency, *OR* Odds Ratio of the effect allele. Effect allele indicates whether the alternative (Alt) or reference (Ref) ERV allele was tested in the model. "Up" or "Down" denotes whether the ERV is associated with an increased ("Up") or decreased ("Down") likelihood of developing leukemia. *P*-values were obtained from a two-sided Wald test and are shown without multiple-testing correction. Closest gene(s) within 10 kb are shown.

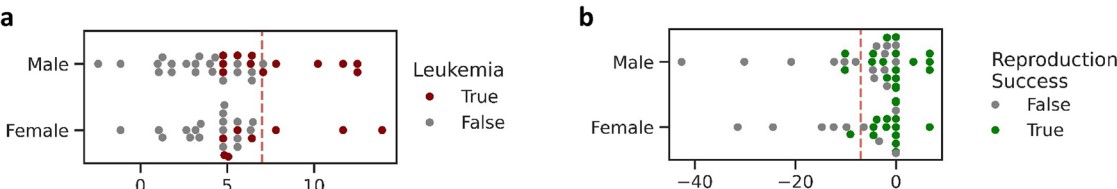

**Fig. 6 | Genetic risk scores. a** Leukemia risk scores. **b** Reproduction success scores. Scores were estimated for deceased koalas based on the odds ratio of the suggestive ERVs associated with cases of leukemia and offspring. Source data are provided as a Source Data file.

The genetic risk score for living animals was also calculated. Based on the score, eight animals were identified as having high risk for developing leukemia. Similarly, four animals from the breeding program had low scores for reproductive success. An overlap of one individual was observed between the two traits analyzed. During the time this study took place, one individual (SB 451) determined as high risk died of leukemia.

In the case of age at death, since it is a continuous trait rather than a case-control variable, a simple breeding index was calculated, similarly to the genetic risk score (Supplementary Fig. 12). This breeding index for longevity should result in animals that could potentially live longer and remain healthier. This should be used together with breeding strategies to mantain genetic diversity, as a tool to support developing a healthier captive population.

## Discussion

This study is to our knowledge the first comprehensive characterization of enKoRV, phaCin-β, phaCin-β-like and SNPs in a large koala pedigree study. The availability of preserved tissue samples, including one animal from the SDZWA founder population, combined with pedigree and health information, makes the current dataset unique and one of the most complete examples for an endangered species.

The patterns of enKoRV and phaCin-β-like in SDZWA koalas resemble those in QLD and NSW. However, the lower number of phaCin-β per individual koala in SDZWA is notable. This pattern may be attributed to a reduced number of phaCin-β in the SDZWA founding individuals, which were born in the Lone Pine Koala Sanctuary, in Brisbane, located north of the Gold Coast region. For example, in the founder we observed 47 phaCin-β integrations whereas the lowest counts in the wild were also observed in the Gold Coast, ranging from 63 to 71 phaCin-β per individual comparable to previous observations ranging from 58 to 71 phaCin-β per individual[6]. Difference in enKoRV in VIC compared to higher allele frequencies of phaCin-β in VIC relative to North Australia could support the hypothesis that enKoRV spread originally from north to south and phaCin-β from south to north. However, recent analysis have estimated the intronic enKoRV in the *SLC29A1* as the first integration event near Coffs Harbor on the Mid-north coast of NSW, spreading from there to north and south[14]. In addition, the severe genetic bottleneck in the southern populations may have resulted in loss of this allele (and others) in some animals and populations due to genetic founder effects and drift.

Older colonization times were established using revised koala mutation rates for enKoRV, phaCin-β, and phaCin-β-like than previously estimated. The older endogenization processes highlight the extended timeframe available for the allele frequency patterns observed. The enKoRV, being more recent, is less abundant than phaCin-β, whereas phaCin-β-like is largely fixed within the population, predating the emergence of modern koalas[12]. The difference in the number of phaCin-β-like insertions per individual between the current study and previous report[6] is due to methodological differences. Previously, only polymorphic reference ERVs were examined. Since phaCin-β-like insertions are generally fixed and present in the reference koala genome, most phaCin-β-like insertions were not reported by Lillie et al.[6].

The pattern of recent ERVs showing higher degrees of polymorphism, in contrast to older ERVs being fixed or close to fixation has been reported for different species, such as mice[28] and rabbits[29]. The stage of endogenization is reflected in the quantities of ERVs in proto-oncogenes, potentially linked to leukemia cases, as well as in fertility and longevity metrics. The higher loss rates for both enKoRV and phaCin-β compared to the emergence of new germline ERVs indicate that the endogenization process is older than previously thought, with phaCin-β clearly exhibiting much lower generation rates than enKoRV.

The different generational rates between EUZ and SDZWA koala populations may be due to methodological variations and sample sizes. The inverse PCR method in EUZ enhanced KoRV integration detection, possibly skewing results. Distinguishing between enKoRV and somatic KoRV was challenging, potentially leading to

**Table 2 | Associations of ERVs with reproduction success**

| ERV | Contig | Position | AAF | OR | Up or down | p-value | Gene | Region |
|---|---|---|---|---|---|---|---|---|
| enKoRV | JAOEJA010001224.1 | 3,086,484 | 0.04 | 0.03 | Down | 0.01 | - | intergenic |
| phaCin-β | JAOEJA010001255.1 | 9,749,859 | 0.11 | 0.08 | Down | 0.01 | - | intergenic |
| enKoRV | JAOEJA010001068.1 | 18,324,756 | 0.04 | 0.02 | Down | 0.02 | - | intergenic |
| enKoRV | JAOEJA010000825.1 | 31,027,633 | 0.06 | 0.05 | Down | 0.02 | - | intergenic |
| enKoRV | JAOEJA010000937.1 | 36,954,807 | 0.05 | 0.02 | Down | 0.02 | - | intergenic |
| enKoRV | JAOEJA010000913.1 | 1,592,048 | 0.14 | 0.17 | Down | 0.02 | *PPFIBP1* | upstream |
| phaCin-β | JAOEJA010000073.1 | 54,044,707 | 0.03 | 0.02 | Down | 0.02 | - | intergenic |
| enKoRV | JAOEJA010000005.1 | 355,168 | 0.05 | 0.06 | Down | 0.02 | *JAM2, MRPL39* | upstream |
| enKoRV | JAOEJA010000517.1 | 9,231,929 | 0.04 | 32.35 | Up | 0.03 | *RAB13, NUP210, RPS27* | upstream |
| phaCin-β | JAOEJA010000944.1 | 4,970,294 | 0.05 | 0.10 | Down | 0.03 | *Vmn2r65* | upstream |
| enKoRV | JAOEJA010000659.1 | 19,889,096 | 0.08 | 0.11 | Down | 0.03 | - | intergenic |
| enKoRV | JAOEJA010000741.1 | 35,843,421 | 0.06 | 0.11 | Down | 0.03 | Two novel genes and *ALDH1A3* | downstream |
| enKoRV | JAOEJA010000741.1 | 23,466,054 | 0.05 | 0.11 | Down | 0.04 | - | intergenic |
| enKoRV | JAOEJA010000548.1 | 25,140,727 | 0.05 | 0.04 | Down | 0.04 | - | intergenic |
| enKoRV | JAOEJA010000889.1 | 6373,854 | 0.05 | 0.08 | Down | 0.04 | - | intergenic |
| enKoRV | JAOEJA010000517.1 | 15,773,814 | 0.08 | 0.15 | Down | 0.04 | *IGSF9* | intron |
| enKoRV | JAOEJA010000824.1 | 30,792,507 | 0.07 | 0.13 | Down | 0.04 | Two novel genes and *FAM71B* | intron and upstream |
| phaCin-β | JAOEJA010000825.1 | 40,974,068 | 0.04 | 25.38 | Up | 0.05 | - | intergenic |

*ERV* Endogenous retrovirus, *AAF* Alternative allele frequency, *OR* Odds Ratio of the effect allele. Effect allele indicates whether the alternative (Alt) or reference (Ref) ERV allele was tested in the model. "Up" or "Down" denotes whether the ERV is associated with an increased ("Up") or decreased ("Down") likelihood of reproduction success. P-values were obtained from a two-sided Wald test and are shown without multiple-testing correction. Closest gene(s) within 10 kb are shown.

overestimation in EUZ. Despite these issues, the identification of new ERVs in both datasets, validated as transmissible to the F2 generation, highlights the dynamic nature of ERV integration.

The high rates of new ERV integrations are consistent with previous observations of an active ERV element in cattle[30]. Although the transmission rate was not reported, the ERVK[2-1-LTR] element in cattle exhibited an average mobilization rate of 1 de novo ERV per approximately 150 sperm cells, with more than a 10-fold difference in rates between individual animals. The de novo ERVK[2-1-LTR] elements tended to preferentially insert near telomeric ends, in GC-rich regions, and within genes. The rates of new integrations and loss of ERV alleles is evolutionary complex and can also result from "competition" with other transposable elements (TEs), likely the case of older ERVs, such as phaCin-β. For instance, it has been proposed that ERVs outcompeted long interspersed nuclear element (LINE) retrotransposons in deer mice[31], being more often observed than LINEs, which are normally the most abundant of the TEs. Another study in the *Mus* lineage suggests that TEs which affect gene expression are rapidly purged. However, most TEs did not change gene expression[28]. Moreover, the same study observed that only a small fraction of TEs and ERVs were responsible for large effects on gene expression[28]. This could be a reason why only the genes *SLC29A1, LONP2*, and *STAP1* containing an ERV were differentially expressed in the 11 koalas analyzed with RNA-Seq data.

The association between the enKoRV in *BCL2L1*[10], eliminated from the captive population, and leukemia, provides an example of a potential purging event of a deleterious ERV. While the distinction between genetic drift and negative selection is difficult, the high OR of a 21-fold increase in leukemia risk supports the hypothesis of negative selection. This was also a very recent event, since the last koala observed with this enKoRV died in 2021. As previously described using long-read sequencing data, the enKoRV in *BCL2L1* gene contained a transduced sequence of *BCL2L1* long-isoform (Bcl-XL) exons 1 and 2[10]. An overexpression of this isoform, which functions as an apoptotic inhibitor[32], could explain the cause of death in the four koalas by

neoplasia, which reinforces the negative effects it had on the health of the individual carriers of this ERV.

Aside from the enKoRV in *BCL2L1*, this study highlights the importance of the enKoRV inserted in the intron of *SLC29A1* gene, given its high frequency across koala populations and its potential link with leukemia (4.9-fold OR for the reference allele) and upregulation in individuals containing the ERV. The potential beneficial impact of this intronic enKoRV could explain its high frequency. Further research on the function of this gene and the effect enKoRV has on it is necessary. Notably, *SLC29A1* gene expression is higher in northern koalas[10,21], when compared to southern individuals lacking the enKoRV, underscoring its importance in wild populations and disease prevalence. A human study has linked this gene to resistance against Azacitidine, a chemotherapy drug[33], which could indicate a role in cancer progression.

Other notable genes associated with ERVs are *LZTS1, LONP2*, and *STAP1*. The enKoRV downstream *LZTS1* showed an OR of 9.3-fold for leukemia, suggesting the importance of this known oncogene for leukemia development in koalas. *LONP2*, while potentially linked to cancer[22], may also affect fertility, as observed in our analysis for reproduction success. While the role of *LONP2* in human female fertility and aging is not clear, there is a clear link between human fertility and *LONP1*[34,35], which could indicate a potential influence of *LONP2* on koala fertility as well. Both *LONP2* and *STAP1* were upregulated in the EUZ population, where their prevalence are considerably higher than in SDZWA. This suggests a need to breed EUZ koalas with individuals lacking these integrations to mitigate the influence of SDZWA genetics in EUZ.

Considering that the process of retroviral endogenization has been occurring in koalas for at least the last 6.9 Mya, and that this is a natural event responsible for shaping 5–10% of mammalian genomes, it does not immediately warrant intervention in the wild. However, when other threats—such as climate change and habitat loss caused by human activity—are added to the equation, the situation worsens. Therefore, monitoring is recommended to allow genetic drift and

natural selection to proceed. Following this, further evaluation of whether intervention is possible or practical should be conducted, especially for critically endangered populations. Nonetheless, for captive populations that serve as gene reserves for the species, it becomes crucial to prevent an increase in the frequency of deleterious ERVs. The aim of captive wildlife breeding should be to produce healthy genetic reserves for future translocations or genetic rescue of wild populations. Current breeding practices in captivity rely on a mean kinship-based strategy that aims to retain gene diversity through equalizing founder representations while also attempting to limit inbreeding. Incorporating individual genetic risk scores based on molecular data can further improve breeding pair selection by supporting breeding among individuals that are at lower risk of developing neoplasia associated with retroviral integrations. However, breeding strategies must also address the demographic needs of the population, and excluding a large number of individuals for high genetic risk scores may reduce the number of potential breeding individuals and lower gene diversity. Future breeding strategies should balance these various genetic and demographic factors. As an example potential application, our genetic risk score predicted one koala to be at high risk for developing leukemia (SB 451), and the individual died of leukemia in December 2024. Excluding this individual from breeding could potentially prevent enKoRVs associated with high risk for neoplasia from being passed to subsequent generations. Further research could potentially aid in managed wild populations in the future. The use of genetic risk scores for breeding success is also something that could be more widely used in captive breeding programs for other at risk wildlife species.

## Methods

### Sample collection
In total, 186 koala individuals were included in this study: 91 from SDZWA, 20 from EUZ, and 75 wild koalas. Blood from living koalas from SDZWA were sampled in accordance with the Institutional Animal Care and Use Committee (IACUC) under IACUC proposals 15-017, 18-024 and 21-019 "Opportunistic Sample Collection during Veterinary Procedures". Samples from deceased animals were taken opportunistically upon death of the animal and stored in the frozen tissues collection of the SDZWA. The 91 koalas from SDZWA (Supplementary Data 1) were shotgun sequenced with Illumina 150 bp paired-end reads at the Illumina laboratory Services. All 91 SDZWA individuals were represented by one sample except three individuals (SB 153, 169, and 308) which had duplicate samples. Sample types included blood ($n = 56$), liver ($n = 23$), lung ($n = 4$), lymph nodes ($n = 4$), nasal mass ($n = 3$), spleen ($n = 3$), and intestine ($n = 1$). Samples were collected between 1992 to 2023 and preserved frozen at -80 °C until DNA extraction. DNA was extracted using the Qiagen QIAamp DNA mini kit blood and tissue protocol. Samples with concentrations greater than 1 mg/μL were diluted using AE buffer. DNA quantitation was performed using a Qubit 4 fluorometer and Qubit dsDNA BR assay kit (Thermo-Fisher Scientific, Waltham, MA, USA) and Quant-iT PicoGreen assay (ThermoFisher Scientific, Waltham, MA, USA). Extracted DNA was kept at −20 °C. Sample type and location are listed in Supplementary Data 1. Complete pedigree information was available for all SDZWA individuals, including cause of death for 58 deceased koalas. From the 58 deceased koala individuals, 27 were neoplasia related mortalities, including lymphoma, leukemia, and lymphosarcoma ($n = 20$), sarcoma ($n = 3$), osteochondroma and osteoma ($n = 3$), and adenocarcinoma ($n = 1$).

In addition, blood was taken from 20 live koalas in the European collections during regular veterinary care, according to each national regulations and institution code of ethic, or opportunistically upon the death of an individual koala. The collections were conducted upon request of the coordinator (Kerstin Ternes) and the veterinary advisor (Baptiste Mulot) of the European Association of Zoos and Aquaria

(EAZA) Ex Situ Program (EEP), in compliance with EAZA ethical standards. Samples were frozen immediately at -20 °C, transported on dry ice and then stored at -80 °C. The DNA was extracted from 100 μl of blood sample using the DNeasy Blood and Tissue Kit (Qiagen, Hilden, NRW, Germany) following the non-nucleated blood cell protocol. The DNA concentration and quality was then determined using an Agilent Tapestation (Agilent Technologies, Santa Clara, CA, USA) with Genomic ScreenTapes and reagents. Koalas from EUZ were sequenced using a PacBio based inverse PCR protocol as described previously[10,36].

Shotgun Illumina data for 75 koalas from wild populations in VIC ($n = 25$), QLD ($n = 25$), and NSW ($n = 25$) was obtained from the open access Koala Genome Survey and scientific and ethics documentation from that project are listed within Hogg et al. [37].

### Paired-end library preparation and sequencing – SDZWA koalas
Paired-end (PE) libraries were generated for the 94 SDZWA samples (91 koalas, including three duplicates) using the TruSeq DNA PCR-Free Sample Prep kit (Illumina, San Diego, CA, USA). Prior to fragmentation, gDNA was purified using paramagnetic sample purification beads (Beckman Colter). DNA was then fragmented and libraries were end-repaired prior to size selection using TruSeq DNA PCR-Free beads for even coverage of areas that are traditionally difficult to sequence. The libraries yield was quantified using a real-time qPCR assay, and Illumina DNA Standards with primer master mix qPCR kit (KAPA Biosystems, Roche, Basel, Switzerland). Following library quantitation, 94 DNA libraries were denatured and normalized to 1.5 nM prior to be being loaded on a NovaSeq6000 instrument where onboard clustering and sequencing occurs on 151 bp paired-end run, generating a minimum of 30X genome coverage per sample.

### ERV Detection of PE reads–SDZWA and wild koalas
Reads were deduplicated, adapters removed, and filtered for an average quality score of 20 using fastp v0.22.0[38]. Bases on the read tails were trimmed if their quality was below 20, retaining only reads with a minimum length of 35 bp. Subsequently, the processed reads were aligned to all KoRV strains (AB721500.1, KC779547.1, KP792564.1, KX587950.1, KU533853.1, KU533852.1, KX587961.1, KX587979.1, KX587966.1, and AB822553.1), phaCin-β, and phaCin-β-like[2] sequences using BWA-mem v2-2.2.1[39]. Unmapped reads with a mate aligned to a viral sequence were retrieved using samtools v1.7[40]. Moreover, soft-clipped regions (exceeding 20 bp) from reads only partially aligned to a viral sequence were extracted, preserving the ERV locations. Due to short length of PE reads, an accurate differentiation between KoRV strains was not possible. Therefore, all reads aligning to a KoRV strain were treated and named KoRV in further analyses; 98.2 and 95.6% of the reads aligning to KoRV in the koalas from SDZWA and in the wild koalas, respectively, aligned to KoRV-A, while the remaining reads aligned to KoRV-B. Clusters of reads containing both KoRV-A and KoRV-B were present in the dataset, which probably indicates incorrect strain assignment.

Both unmapped and soft-clipped reads were subsequently aligned to a retrovirus-masked koala assembly (accession number JAOEJA000000000)[41] using BWA-mem v2-2.2.1[39], with alignments filtered to include only those with a minimum mapping quality score of 30. The masking was performed using RepeatMasker v4.1.2-p1[42], for the same viral sequences used in the first alignment step. Furthermore, ERV locations within the koala genome were identified by clustering at least 20 reads, requiring a maximum inter-cluster read-position difference of 500 bp for KoRV and 9 kbp for other viruses – since the JAOEJA000000000 assembly is enKoRV-free, but not free of other ERVs, which are normally around 8 kbp long. This approach suggests the proximity of an ERV based on the alignment between reads in the koala genome and their respective read-pair mates aligned to the retrovirus (Supplementary Fig. 11).

The exact ERV positions were determined based on the prevalence of overlaps among soft-clipped reads, distinguishing between novel and known integrations. Novel integrations were defined as those not identified in the JAOEJA000000000 assembly. This is the case for all KoRV integrations, as this assembly originates from an enKoRV-free southern Australian koala. The classification into novel or known integrations was based on the analysis conducted by Repeat-Masker, with novel integrations pinpointing to a single exact position. In contrast, known integrations previously identified in the JAOEJA000000000 assembly were characterized by both start and end positions.

In total, before filtering, 9178 KoRV, 9040 phaCin-β, 83601 phaCin-β-like, were detected for all 94 koala samples. ERVs were considered shared between individuals when their cluster locations overlapped, based on the start and end positions. A final list of shared ERVs was used to call zygosity.

## ERV zygosity

The zygosity of the integrations was estimated after initially aligning all trimmed reads to the unmasked koala JAOEJA000000000 assembly. This estimation was based on comparing the distribution of soft-clipped reads against reads spanning the ERV positions. Reads at these positions were deemed soft-clipped if they were truncated by BWA and terminated precisely at, or within 10 bp of, the ERV positions—accounting for minor mismatches due to the Target Site Duplication (TSD) resulting from the proviral integration mechanism. Conversely, reads were classified as crossing if they spanned the integration sites, with start and end points situated beyond the TSD margins (Supplementary Fig. 12).

Zygosity was assessed differently for novel insertions compared to those previously observed in the JAOEJA000000000 assembly. For novel variants, a ratio threshold of 30% of the lesser-read group count was applied to classify insertions as heterozygous. An insertion was deemed homozygous if over 70% of the pertinent reads were soft-clipped; otherwise, an ERV with less than 30% soft-clipped reads was categorized as having undetermined zygosity, possibly indicating false positives or somatic insertions. For known insertions, both start and end positions were used for classification, with homozygosity determined by over 70% crossing reads.

At the population level, any integration not detected was assumed homozygous reference (indicating the absence of an integration). For this study, the viral integration represents the alternative allele, while its absence signifies the reference allele. In addition, the Mendelian error rate was calculated for all available triads or pair parent-offspring. In total, 63 koalas had at least one parent genotyped. AAF, percentage of undetermined zygosity, and Mendelian error per ERV were used for filtering. Only ERVs with an AAF > 0, percentage of undetermined zygosity ≤ 0.1, and Mendelian error rate per ERV ≤ 0.05 were kept for further analyses. An average Mendelian error rate of $0.1 \pm 0.77\%$ was observed per animal after filtering. In addition, genotype concordance was also checked for the three replicates, ranging from 96.0 to 98.0%. For subsequent analyses, a consensus between the replicates of each of the three koalas was used, being inconsistent genotypes coded as undetermined (coded as -1).

## SNP calling of PE data

Reads filtered for quality control as previously described in the ERV detection step were aligned to the koala assembly JAOEJA000000000 using dragmap v1.3.0[43], and sorted and merged with samtools v1.7[40]. SNPs and indels were called using GATK v4.4.0[44] per sample on *HaplotypeCaller* tool and then recalled population wide on *GenomicsDBImport* and *GenotypeGVCFs* tools. Finally, SNPs and indels were filtered with the *VariantFiltration* tool of GATK using several criteria. Variants with an RMS Mapping Quality (MQ) < 40.0, a Fisher Strand Bias (FS) > 60.0, a Quality by Depth (QD) < 2.0, a Strand Odds Ratio (SOR) > 3.0, a Mapping Quality Rank Sum Test (MQRankSum) < -12.5, a Read Position Rank Sum Test (ReadPosRankSum) < -8.0, and a Depth (DP) < 10 were removed from the analysis. In addition, only variants with a call rate > 90%, totalizing 21,083,491 SNPs and indels were used for further analysis.

Furthermore, SNPs for enKoRV were called similarly to the host SNPs, using the mapped reads to KoRV-A reference sequence (AB721500). Filtering of reads followed the same thresholds, totaling 111 SNPs.

## iPCR and PacBio sequencing−EUZ koalas

DNA was sheared and circularized[10], for the 20 samples from the EUZ. Sonication inverse PCR (SIP) reactions were then performed on all samples (fragmented and circularized) using the following LTR primers: forward 5′-ATTTGCATCCGGAGTTGGT-3′ and reverse 5′-AGGGGCACCCTAGAAACTGT-3′. Libraries were then created for each sample using the PacBio (Pacific Biosciences, Menlo Park, CA, USA) 5 kb template preparation protocol and the SMRTbell™ Template Prep Kit 1.0 following the manufacturer's guidelines. Sequencing on the PacBio Sequel II platform was performed using the MagBead Standard protocol, C4 chemistry and P6 polymerase on a single v3 Single-Molecule Real-Time (SMRT) cell with 1 × 180 min movie for each sample. Every sample was run across two Single-Molecule Real-Time (SMRT) cells to increase the coverage. The reads from the insert sequence were processed within the SMRT®Portal browser (minimum full pass = 1; and a Minimum Predicted Accuracy of 90).

## enKoRV detection of long-reads−EUZ koalas

KoRV integrations were detected as previously described[10]. Briefly, parts of the PacBio reads mapping to any KoRV strains were identified using BLAST v2.12.0[45] and removed so that only the koala genomic sequence (flanking regions of the provirus) remained. The trimmed reads were aligned to the South Australian koala genome (GCA_030178435.1) using Minimap v2.1[46] without secondary alignments. Matches longer than 50 bp, with ≥ 90 % of matching bases, and with a mapping quality ≥ 30 were retained. KoRV integrations were detected by finding peaks of alignment coverage >10 reads per breakpoint with an overlap within 10 bp (as of the TSD) between the breakpoints, except for cases of masked sequences, where the maximum masked sequenced for recKoRV of 1,411 bp was allowed between breakpoints. ERV and somatic KoRV were defined based on coverage, with a cutoff of 500 reads established as the minimum threshold to define enKoRV. An exception was made for one koala (SB 369), where the threshold was set at 1000 reads (see Supplementary Fig. 13).

## RNA-Seq and visualization of ERVs

RNA-Seq data from five publicly available KoRV-free koala individuals was used (PRJEB21505) in addition to RNA-Seq from six of the EUZ koalas sequenced with PacBio. RNA-Seq data was processed following the open-source snakemake pipeline[47], trimming reads with fastqc v0.12.1[48], aligning and sorting reads with hisat2 v2.2.1[49] and samtools v1.17[40], respectively, and counting reads with htseq-count v2.0.8[50]. Gene count was normalized using the DESeq2 v4.1.3 package in R v4.1.1. For each of the genes found with an ERV in our study and present in the RNA-Seq two groups were formed, one with the ERV and the other without it. The two groups were compared with a two-side Mann-Whitney test using the SciPy v1.11.4 package[51] in Python v.3.12.

ERVs were visualized using either the reference genome sequence, or alignment of assembled reads in the Geneious Prime v2025.0.2 software (www.geneious.com).

## PCA

Principal component analysis (PCA) of both SNP and ERV data were performed using the Scikit-learn v.1.5[52] package in Python v.3.12. In the

case of enKoRV data alone, data was first transformed using standard scale. In addition, only non-missing markers with an AAF ≥ 0.05 were used.

### Detection of joey specific ERVs

New ERV integrations were assessed in 55 triads, 46 triads from the SDZWA population and nine from the EUZ population. In the case of SDZWA triads, new ERV integrations were represented by heterozygous high-coverage ERVs unique to the joey and absent in both parents (coverage ≥ 40 reads, corresponding to a 0.6% probability of missing heterozygous ERVs based on a binomial model for a 30–70% heterozygous ratio). New ERVs with lower coverage were very likely somatic integrations detected. Since 19 F2 were available for those triads, new ERVs could be validated in the next generation. For these, the presence of one copy of the ERV in the offspring with a coverage ≥ 10 reads was used as validation. In the case of EUZ, only KoRV integrations were available, and at much higher coverage, given the inverse PCR methodology used[15]. The same approach used for the detection of new ERVs in the SDZWA was used for EUZ koalas, except for the zygosity estimation and the definition of somatic integrations. In order to remove somatic integrations enriched, a coverage threshold at ≥ 500 reads for both breakpoints of an integration was established, keeping only integrations with a coverage above this threshold. The coverage for the joey SB 369 was set to 1000 reads due to higher coverage rate (manually curated based on the coverage distribution of unique versus shared ERVs – Supplementary Fig. 13).

### Mutation rate and initial germline colonization estimates

The germline mutation rate in koala was estimated based on the 46 triads from the SDZWA. High-quality unique SNPs not detected in either parent were considered a de novo mutation, using GATK v4.4[44] (*FindMendelianViolations*). A more stringent quality control for de novo SNPs was applied based on recommendations of the results from the Mutationathon benchmark[53]. Those include a minimum depth of 20 reads, minimum genotyping quality of 20, and minimum allele balance of 0.4 at sites that are heterozygous in the offspring. Mutation rate per base pair per generation (mutations_site_generation) was calculated as the total number of de novo SNPs divided by the number of triads multiplied by the diploid assembly length (6,469,964,576 bp). Mutation rate per base pair per year (mutations_site_year) was calculated in the same way, except that the total number of SNPs was estimated per year, summing up the number of detected SNPs per year per joey divided by the arithmetic mean age of the parents.

$$\text{mutations\_site\_generation} = \frac{\sum_{k=0}^{n}\text{denovoSNPs}}{n_{\text{trios}}*\text{assembly}_{\text{length}}*2} \quad (1)$$

$$\text{mutations\_site\_year} = \frac{\sum_{k=0}^{n}\frac{\text{denovoSNPs}}{\frac{(\text{Age}_{\text{sire}}+\text{Age}_{\text{dam}})}{2}}}{n_{\text{trios}}*\text{assembly}_{\text{length}}*2} \quad (2)$$

The initial germline colonization times of KoRV, phaCin-β and phaCin-β-like were estimated based on the 5'-LTR and 3'-LTR divergence within the same provirus. Since the 5'-LTR and 3'LTR are identical at the time of integration, any variations accumulated overtime work as a molecular clock that can be used to estimated earliest time of integration. The enKoRV located in the intron of the *SLC29A1* gene was used to determine the LTR divergence, since it is the enKoRV with the highest frequency in the koala population and very likely one of the most ancient, which shows divergence (p-distance) of 0.001 (one insertion/deletion observed in 1000 bases - Supplementary Fig. 4). In the same way, germline colonization times of phaCin-β and phaCin-β-like were updated based on the median LTRs divergence previously reported[2], following the formula[13]:

$$\text{Colonization time} = \frac{\text{LTR divergence}}{\text{mutations\_site\_year}*2} \quad (3)$$

### Genome-wide association

Using a logistic regression in PLINK v2, SNPs and ERVs were analyzed in a case-control model for leukemia, leukemia strict control (without other cases of neoplasia as controls), neoplasia, neoplasia without leukemia, and neoplasia without leukemia strict control, and fertility (in terms of total offspring: 24 koalas with no offspring versus 34 with offspring ranging from 1 to 14). In addition, ERVs and SNPs were tested for longevity determined as the age at death using a linear regression in PLINK v2. Markers with Minor Allele Frequency (MAF) ≥ 0.01 were tested for the additive effects. AIC (Akaike information criterion) was used to test which available cofactors including age at death, total offspring, sex, death location, birth location, cause of death, and PC1–PC3 were contributing to each regression model (DeltaAIC < -5).The cofactor total offspring was used in the longevity model, while the cofactor age was used in the fertility model. For the other investigated phenotypes, no cofactor contributed to the models.

For ERVs and SNPs, thresholds for significant ($p < 0.05$) associations were estimated using Bonferroni multiple testing correction considering 953,591 independent SNPs derived from linkage disequilibrium based pruning in PLINK v2 with the parameters window size=500, step size=100, and $r^2 = 0.5$. A log base 10 threshold of 1.3 ($p < 0.05$ without multiple testing correction) was used to detect suggestive ERVs and of 6 for suggestive SNPs. Genes were annotated for associated SNPs and ERVs, and their flanking region of ±10 kb, using previously predicted gene annotation[41]. Scaffolds were assigned to chromosomes as defined in Supplementary Note 1. ERV and SNP consequences were predicted using Ensembl Variant Effect Predictor release 105[54]. Association to viral SNPs were also tested as described in Supplementary Note 2, Supplementary Fig. 15, and Supplementary Data 9.

### Genetic risk score

Genetic risk score (GRS) and the longevity breeding index were estimated to assess genetic contributions to traits. The GRS was computed using odds ratio (OR) values obtained from suggestive markers associated with leukemia and fertility, where each individual's score was derived by summing the products of these OR coefficients and their respective genotype counts. The longevity breeding index, since it is a continuous trait, similarly utilized beta values.

To estimate prediction accuracy of the GRS estimates, we employed a bootstrapping approach with 100 iterations, randomly splitting the dataset into training (75%) and testing (25%) sets. A logistic regression model was trained on the GRS to predict outcomes. We calculated accuracy as the proportion of correctly classified cases and averaged the results from all iterations, providing a robust estimate of prediction accuracy. Different cutoffs of the GRS were tested, and the cutoff which generated the highest accuracy was selected. The GRS calculated for living animals was used then to detect animals falling into the cutoff estimates, as individuals one would recommend to be avoided in SDZWA koala breeding program.

### Reporting summary

Further information on research design is available in the Nature Portfolio Reporting Summary linked to this article.

## Data availability

Publicly available data from the Koala Genome Survey (https://awgg-lab.github.io/australasiangenomes/species/Phascolarctos_cinereus.

html) and RNA-Seq data from five KoRV-free koala individuals was used (European Nucleotide Archive, project PRJEB21505). Data generated in the current work are deposited at European Nucleotide Archive, under the project PRJEB86269. Source data are provided with this paper.

## Code availability

Code available at GitHub[55]: https://github.com/neumannguib/Koala_SDZWA.

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

## Acknowledgements

The authors would like to acknowledge Illumina Inc. for sequencing the SDZWA koalas and the support from the Illumina Laboratory Services sequencing team. The authors thank Max Coenen for sharing scripts, Karin Hönig and Nick Peters for technical support throughout the project, Dr. Gayle McEwen for methodological insights provided on the analysis of EUZ data, and Asako Chaille for providing information and advice about the koala breeding strategy of the SDZWA. The authors gratefully thank the veterinary and animal collection staff at the following institutions: ZooParc de Beauval, Saint-Aignan, France; Jardim Zoológico, Lisbon, Portugal; Zoo Aquarium de Madrid, Madrid, Spain; The Royal Zoological Society of Scotland, Edinburgh, UK; Zoo Planckendael and Zoo Antwerp, Belgium; Zoo Duisburg GmbH, Duisburg, Germany; and Tiergarten Schönbrunn, Vienna, Austria. Open Access funding enabled and organized by Projekt DEAL. ADG was supported by grant GR 3924/15-1 from the Deutsche Forschungsgemeinschaft (DFG). DEA-P and ADG were supported by the Morris Animal Foundation, Grant D14ZO-94. PJ was supported by the Swedish research council VR grant 2024-03215. ADG and ALR were supported by Grant R01GM092706 from the National Institute of General Medical Sciences (NIGMS). The content is solely the responsibility of the authors and does not necessarily represent the official views of the NIGMS or the National Institutes of Health.

## Author contributions

A.D.G., R.T., C.S. and G.B.N. conceived and designed the study. G.B.N. performed all analyses. P.K. performed the genome-wide association analysis. A.D. contributed with the laboratory work done for the EUZ koalas, while P.M.G. and C.S. contributed with the laboratory work done for the SDZWA koalas. K.A.V.-M., S.U. and K.N. coordinated the sequencing of the SDZWA koalas. B.M. is the European Koala Population Vet Advisor and oversaw sample collection and pedigree information for EUZ koalas, while C.S. collected pedigree information for SDZWA koalas. A.L.R., P.J., K.T. and D.E.A.-P. provided advice and helped to write the manuscript. G.B.N., R.T. and A.D.G. wrote the manuscript. All authors reviewed and approved the final version.

## Funding

## Competing interests

The authors declare no competing interests.
