## [Transparent Peer Review file · Nature Communications]

Multi-generational koala pedigree analysis reveals rapid changes in heritable provirus load associated with life history traits

Corresponding Author: Professor Alex Greenwood

Version 0:

Reviewer comments:

Reviewer #1

(Remarks to the Author)

This paper reports a detailed analysis of the endogenous retroviruses found in koalas. It is novel and interesting but very hard going because of the vast amount of data reported. I can't help wondering whether it might have been better to focus on the enKoRVs, leaving a comparison with the other elements for a follow up paper. Certainly, it would be easier on the reader and I doubt whether the main conclusions would change. In any event it is an impressive body of work.

Some specific comments

L57. Insert 'per base pair per year' after 'of humans'

L93-5. Relatedness to what?

L116. (ST1) Why are entries to the Studbook not listed in numerical order?

L127. The number of phaCin- β -like elements is much higher than previous estimates (L71-6. Some comment would appear indicated.

L162. If the phaCin- β and phaCin- β -like colonization dates increase by c25% why does that of the enKoRV elements increase several fold? Is there overreliance on the SLC29A1 provirus?

L180. Are there really numerous segregating phaCin- β -like elements?

L217. Isn't it surprising that neither parent of SB153 shows this relationship?

L222. There seems to have been some confusion in quoting ref 14 ("Given the historical population bottleneck of Narrandera koalas in the 1890s and their subsequent reintroduction from Queensland and Victoria in the 1970s, we designated....")

L269. Is there no information on the cause of death of SB21, 24, 111, 196?

L278. Or seven? Does SB268 not count?

L273-284. I wonder what conclusion on is supposed to draw from this section given that the provirus clearly has been transmitted to offspring? Is there any evidence for fetal malformations or other harm in koalas?

L484. Given that these animals have a decent lifespan how would such negative selection work? Reduced fertility?

L531. One might argue that the first priority should be to try to identify individuals that show no evidence of reinsertions.

L541. How similar was the proviral content of duplicate samples from the same individual?

(Remarks on code availability)

Reviewer #2

(Remarks to the Author)

In this manuscript, analysis of the endogenous retroviruses (ERVs) in the Koala was carried out. Virtually all higher organisms carry multiple copies of ERVs that result from germline infection (and insertion of viral DNA), followed by transmission of the ERVs to subsequent generations. The majority of ERVs result from ancient infections in progenitors to the modern day species, and the great majority of ERVs have become fixed in the genomes. The retroviruses that gave rise

to the ERVs are generally no longer present as infectious agents for most species. The koala is a unique case in that it carries two infectious retroviruses, KoRV and phaCin-beta that are currently generating new ERVs. This is evident from the facts that many ERVs of these two classes are not fixed in the koala genome, and that new ERVs (particularly endogenous KoRVs) can be detected in offspring. Study of the ERVs in koalas provides a window into early phases of ERV endogenization which cannot be studied in other species (including humans).

This report continues this group's studies of koala retroviruses and ERVs, focusing on koala colonies from European zoos, and from the San Diego Zoo Alliance. They combine a broad range of techniques, including genomic sequencing and RNAseq, with family pedigrees (over as many as 4 generations) and health records of 91 animals. They have identified by DNA sequencing more than 2000 ERVs in the population. The great majority of the endogenous KoRV and phaCin-beta ERVs were not fixed in the genome, while an older phaCin-beta-like ERV was largely fixed. The investigators applied population genetic analyses which provided insight into the impacts of new koala ERVs on risk for neoplasia and reproduction.

This study is a powerful combination of genetic and molecular studies with genealogy and health records in two well-studied populations that provides important new insights into the effects of ERVs as they invade a species. Implications are clear for our general understanding of ERVs of other species such as humans where ERVs make up ~8% of the genome. The first studies in the paper recalibrating when KoRV and phaCin-beta entered the germ lines of koalas or their ancestors, as well as characterization of the distribution of ERVs in the zoo populations compared to wild koalas in northern and southern Australia are convincing.

1. In the section on ERVs through the generations and the examples in Figs 3 and 4, the implication is that the ERVs are responsible for the failure of the progeny to survive or breed. One possibility could be that these ERVs are producing infectious KoRV which could lead to the neoplasia or loss of fertility. Do animals with these inherited ERVs have higher KoRV viral loads? Perhaps future RNAseq or qRT-PCR assays could shed light on this. For the pedigree in Fig 3, if tumor tissues are available, were there somatic KoRV integrations near other proto-oncogenes that might collaborate with the BCL2L1 insertion? Also, since the ERVs in both Fig 3 and Fig 4 were originally present in more than one animal, a block to genetic transmission was obviously not absolute.
2. One of the most interesting results from the study are the quite high rates of new endogenizations observed over just one generation (i.e. the triads analyzed). However, the results for the SDZWA and EUZ analyzed should be reported similarly. For EUZ, 4 of the 8 triads showed new ERVs; for the SDZWA animals the results are reported in terms of the number of new ERVs (16) in the 46 triads, but it appears that these new ERVs were confined to 8 of the 46 triads. In any event, the high frequency (and frequent instances of more than one new ERV in an F1) bears comment in the Discussion. Is there evidence of high KoRV or phaCin-Beta expression or infection in tissues of the female or male reproductive tract?
3. Another strength of this work is that the effects of koala ERVs (in aggregate as well as individually) on cancer rates, fertility and longevity were determined. As mentioned above, it will be important to distinguish between high levels of infectious KoRV (or phaCin-beta) vs. effects of the proviral insertion on the host gene. The effects of a given ERV may reflect either or both of these processes. The ERVs associated with altered cancer incidence in Table 1 and those associated with reproductive success in Table 2 are interesting. The written narrative (351-413) is impeded by description of the functions of many of the host genes at the insertion sites. It would be clearer to focus on a few ERVs where there is a strong case for biological importance, e.g. the insertions in SLC29A1 and BCL2L1. The other host sites could be grouped into proto-oncogenes, tumor suppressor genes, etc. and simply listed by name. For instance the description of the function of LZTS1 on lines 358-360 could be eliminated.
4. Finally, it would be helpful for the authors to set their studies into a larger biological/evolutionary framework in the Discussion. On the one hand their results demonstrate significant effects of ERVs (particularly KoRVs) on cancer and reproductive capacity on koalas, as well as high ongoing endogenization of new ERVs. On the other hand, they now calculate that KoRV (the most recent koala ERV) may have begun to infect at least some koalas as far back as 300,000 years ago, concomitant with emergence of modern koalas. And KoRV as an infectious agent may be spreading from north to south in Australia. What are the Implications for the wild koala population?

(Remarks on code availability)

Reviewer #3

(Remarks to the Author)

Neumann and team have presented a manuscript that significantly advances our understanding of key retroviruses in the koala. Currently the koala is unique in that it is experiencing germline invasion by two viruses, KoRV and phaCin-B. The study of the process of invasion and inheritance by these viruses provides key understandings to not only the koala but much more broadly. The authors have significantly advanced previously published work by using a powerful set of 111 koala samples from family trees of koalas with up to three known generations. This enabled them to follow inheritance of key insertion sites in the koala genome. The use of these multi-generational pedigrees combined with key health data on both living and deceased koalas enabled the authors to link key insertional sites with health predictors. Finally, the key retroviral insertions could be associated with health signals such as leukemia, fertility and longevity, enabling the authors to develop a genetic risk score, which will be very useful for breeding of koalas in captivity in particular. The authors have presented a significant amount of raw data, both in the main manuscript and also in Supplementary files, and this data strongly supports their study and their conclusions. The science is high quality and the methodology is strong and very well structured.

This study is the first comprehensive analysis of such a large and well structured pedigree to further characterise koala retrovirus integrations and their positive and negative selections.

One minor error on Line 445. The Lone Pine Koala Sanctuary is located in Brisbane, not on the Gold Coast of Queensland Australia.

I strongly support the publication of this manuscript.

(Remarks on code availability)

This is outside my area of expertise

Version 1:

Reviewer comments:

Reviewer #1

(Remarks to the Author)

I am happy to support publication of this manuscript in revised form although I still feel that a degree of simplification by focussing on the KoRVs would be advantageous. However, I do fully understand the authors' wish to include all the available data in one place.

I have a few minor comments/corrections:

L84. Insert 'potentially' before deleterious

L94. More than?

L306. 'Expected' or 'accepted' rather than excepted

L529. Where does the figure of 300,000 come from? The phaCin- (like) elements are clearly older while the KoRV data seems rather soft ("younger estimated dates cannot be discarded"-L166)

(Remarks on code availability)

Reviewer #2

(Remarks to the Author)

In their revised manuscript, the authors have addressed the questions raised in my previous review and modified the presentations where appropriate. I recommend publication of this work. It is comprehensive, novel and appropriate for this journal.

(Remarks on code availability)

RESPONSE TO REVIEWERS' COMMENTS

Reviewer #1 (Remarks to the Author):

This paper reports a detailed analysis of the endogenous retroviruses found in koalas. It is novel and interesting but very hard going because of the vast amount of data reported. I can't help wondering whether it might have been better to focus on the enKoRVs, leaving a comparison with the other elements for a follow up paper. Certainly, it would be easier on the reader and I doubt whether the main conclusions would change. In any event it is an impressive body of work.

Response: We thank the reviewer for their thorough and positive assessment of the work. We do understand the concern about the amount of data, but the authors decided to report it all together as it contributes to the same story line and strongly support the results. In particular, the genetic risk scores can be used to implement changes in captive koala (and perhaps at some point managed wild koala) breeding programs. Therefore, it is important they have all available data to support their decisions.

Some specific comments

L57. Insert 'per base pair per year' after 'of humans' -

Response: Done. Placed before humans, since it is the case for both humans and mice

L93-5. Relatedness to what? –

Response: we changed to “Comparisons with wild koalas indicated that the SDZWA koalas are genetically more closely related to northern Australian koalas. In addition, there is evident genetic relatedness between SDZWA koalas and koalas from European zoos as expected due to translocations between zoos in North America and Europe.”

L116. (ST1) Why are entries to the Studbook not listed in numerical order? –

Response: We thank the reviewer for noticing this error in presentation. We changed to numerical order.

L127. The number of phaCin- β -like elements is much higher than previous estimates (L71-6. Some comment would appear indicated.-

Response: The numbers reported previously were only for polymorphic phaCin- β -like, while in our study we report all integrations, including fixed phaCin- β -like. This has to do with the difference in methodology. Since phaCin- β -like are fixed and present in the koala reference genome, Lillie et al. only detected deletions, meaning polymorphic reference ERV loci. So that means that our numbers are much smaller. Only six phaCin-

β -like were segregating in SDZWA and 14 when we also consider the 75 wild koalas, out of 361 phaCin- β -like in total. If we would use all 430 koalas, as previously, the numbers would probably increase, out of 469 phaCin- β -like previously detected in total (Lillie et al, 2024). We added to the discussion now the following sentence: L465 “The difference in the number of phaCin- β -like insertions per individual between the current study and previous report (Lillie et al., 2024) is due to methodological differences. Previously, only polymorphic reference ERVs were examined. Since phaCin- β -like insertions are generally fixed and present in the reference koala genome, most phaCin- β -like insertions were not reported by Lillie et al.”.

L162. If the phaCin- β and phaCin- β -like colonization dates increase by c25% why does that of the enKoRV elements increase several fold? Is there overreliance on the SLC29A1 provirus?

Response: The difference in the increase for enKoRV is due to a methodological difference in the calculation. We opted to keep the same approach applied in the study of phaCin- β and phaCin- β -like (Lillie et al., 2024), as referred to in the Methods.

L180. Are there really numerous segregating phaCin- β -like elements? -

Response: We thank the reviewer for the observation. Numerous was not an appropriate term. We decided to remove this statement, keeping only the information about the AAFs of phaCin- β : “However, in comparison to SDZWA, the AAFs of phaCin- β in VIC were higher (Supplementary Figure 7).”.

L217. Isn't it surprising that neither parent of SB153 shows this relationship? –

Response: It is indeed a bit surprising. This was likely caused by the lower sequence coverage of the liver sample. In this case, we had duplicates: one sample from lung tissue and one from liver tissue. Since we decided to use the consensus genotype instead of the genotype from the sample with the highest coverage, the number of ERVs for this individual was reduced to 417, compared to an average of 470 ERVs per koala in the SDZWA population. Nevertheless, the low Mendelian error rates of 0.12% for the lung and 0.04% for the liver validate the relationship within the triad. Additionally, based on the called SNPs, this individual is correctly placed within the SDZWA clade, in direct association with the parents (not reported in the manuscript).

L222. There seems to have been some confusion in quoting ref 14 (“Given the historical population bottleneck of Narrandera koalas in the 1890s and their subsequent reintroduction from Queensland and Victoria in the 1970s, we designated...”) –

Response: We thank the reviewer for catching this mistake. We changed the time range from the 1890s to 1970s.

L269. Is there no information on the cause of death of SB21, 24, 111, 196?

Response: Unfortunately not for all of them. SB21 and SB24 died in Australia. SB111 was apparently healthy, and had anesthetic death (no histology report available). SB196 had a spinal fracture. We color only the death by leukemia. We change the figure caption to "...Death or euthanasia due to leukemia (where known) is indicated by color pattern".

L278. Or seven? Does SB268 not count?

Response: True, it is indeed seven. We did not count SB268, as it was the only observed offspring that inherited the ERV, but considering the way sentence is written, we should also count it.

L273-284. I wonder what conclusion one is supposed to draw from this section given that the provirus clearly has been transmitted to offspring? Is there any evidence for fetal malformations or other harm in koalas?

Response: There is no clear evidence for fetal malformations, although, it is known in koalas. We believe this ERV could be associated with decreased fertility, in other words, an increased predisposition to abortion, for instance. That is not necessarily a mendelian trait, rather a complex one. Based on the graph, we could infer that SB108, SB210, and SB240 probably carried the same ERV. None of them seemed to have struggled to have offspring, having 4, 5, and 6, respectively. Nevertheless, we believe, it is worth the observation, since, other people working with koalas could eventually detect the same ERV.

L484. Given that these animals have a decent lifespan how would such negative selection work? Reduced fertility?

Response: The idea is that the carriers of this ERV die younger, reducing the potential offspring number, and reducing the chances of transmitting the ERV. The average age at death of our cohort was 9 years old. From the four carriers, two females died at ages 5 and 7 years old, with total offspring 3 and 5 joeys each. While, the males lived both 10 years, and had an offspring of 3 and 9 joeys each.

L531. One might argue that the first priority should be to try to identify individuals that show no evidence of reinsertions.

Response: The problem is that the reinsertions only show up during the developing of neoplasia, as previously observed comparing tumor tissues versus health tissues (Gayle et al, 2021). When the tumor is observed, it is already too late and the animal is euthanized.

L541. How similar was the proviral content of duplicate samples from the same individual?

Response: Since no enrichment was performed, most proviruses detected were ERVs. Some somatic viral insertions detected were probably discarded due to low coverage during zygosity calling. The consensus between the duplicates varied from 93% to 96% for raw data, and 96% to 98% after filtering (see Methods, L673).

Reviewer #2 (Remarks to the Author):

In this manuscript, analysis of the endogenous retroviruses (ERVs) in the Koala was carried out. Virtually all higher organisms carry multiple copies of ERVs that result from germline infection (and insertion of viral DNA), followed by transmission of the ERVs to subsequent generations. The majority of ERVs result from ancient infections in progenitors to the modern day species, and the great majority of ERVs have become fixed in the genomes. The retroviruses that gave rise to the ERVs are generally no longer present as infectious agents for most species. The koala is a unique case in that it carries two infectious retroviruses, KoRV and phaCin-beta that are currently generating new ERVs. This is evident from the facts that many ERVs of these two classes are not fixed in the koala genome, and that new ERVs (particularly endogenous KoRVs) can be detected in offspring. Study of the ERVs in koalas provides a window into early phases of ERV endogenization which cannot be studied in other species (including humans). This report continues this group's studies of koala retroviruses and ERVs, focusing on koala colonies from European zoos, and from the San Diego Zoo Alliance. They combine a broad range of techniques, including genomic sequencing and RNAseq, with family pedigrees (over as many as 4 generations) and health records of 91 animals. They have identified by DNA sequencing more than 2000 ERVs in the population. The great majority of the endogenous KoRV and phaCin-beta ERVs were not fixed in the genome, while an older phaCin-beta-like ERV was largely fixed. The investigators applied population genetic analyses which provided insight into the impacts of new koala ERVs on risk for neoplasia and reproduction.

This study is a powerful combination of genetic and molecular studies with genealogy and health records in two well-studied populations that provides important new insights

into the effects of ERVs as they invade a species. Implications are clear for our general understanding of ERVs of other species such as humans where ERVs make up ~8% of the genome. The first studies in the paper recalibrating when KoRV and phaCin-beta entered the germ lines of koalas or their ancestors, as well as characterization of the distribution of ERVs in the zoo populations compared to wild koalas in northern and southern Australia are convincing.

We thank the reviewer for their positive assessment and for providing valuable insights regarding its potential outcomes.

1. In the section on ERVs through the generations and the examples in Figs 3 and 4, the implication is that the ERVs are responsible for the failure of the progeny to survive or breed. One possibility could be that these ERVs are producing infectious KoRV which could lead to the neoplasia or loss of fertility. Do animals with these inherited ERVs have higher KoRV viral loads? Perhaps future RNAseq or qRT-PCR assays could shed light on this. For the pedigree in Fig 3, if tumor tissues are available, were there somatic KoRV integrations near other proto-oncogenes that might collaborate with the BCL2L1 insertion? Also, since the ERVs in both Fig 3 and Fig 4 were originally present in more than one animal, a block to genetic transmission was obviously not absolute.

Response: This is a very interesting idea. Unfortunately, we do not have any data on viral loads for this population. However, we agree that this information could be very informative, and we plan to conduct more RNA sequencing in the future, as tumor tissues from these animals were not available. Nevertheless, there is clear support from our previous work (Gayle et al., 2021) showing reintegrations near other proto-oncogenes in tumor tissues. Checking whether the same regions are enriched in tumor tissues of koalas from SDZWA would be very interesting.

2. One of the most interesting results from the study are the quite high rates of new endogenizations observed over just one generation (i.e. the triads analyzed). However, the results for the SDZWA and EUZ analyzed should be reported similarly. For EUZ, 4 of the 8 triads showed new ERVs; for the SDZWA animals the results are reported in terms of the number of new ERVs (16) in the 46 triads, but it appears that these new ERVs were confined to 8 of the 46 triads. In any event, the high frequency (and frequent instances of more than one new ERV in an F1) bears comment in the Discussion. Is there evidence of high KoRV or phaCin-Beta expression or infection in tissues of the female or male reproductive tract?

Response: We thank the reviewer for the comment. We changed the results to report both results similarly. Indeed, the detection of new ERVs was unequal in both populations. We speculate this might be due to differences in the methods, as data on

enKoRV for EUZ was enriched, causing difficulty in the differentiation between somatic and endogenous insertions. Regarding the high rates, we now added to Discussion “The high rates of new ERV integrations are consistent with previous observations of an active ERV element in cattle (Tang et al., 2024). Although the transmission rate was not reported, the ERVK[2-1-LTR] element in cattle exhibited an average mobilization rate of 1 *de novo* ERV per approximately 150 sperm cells, with more than a 10-fold difference in rates between individual animals. The *de novo* ERVK[2-1-LTR] elements tended to preferentially insert near telomeric ends, in GC-rich regions, and within genes.”. In the future, one could attempt to analyze koala sperm to quantify the rates of new insertional ERV polymorphisms in the male germline.

3. Another strength of this work is that the effects of koala ERVs (in aggregate as well as individually) on cancer rates, fertility and longevity were determined. As mentioned above, it will be important to distinguish between high levels of infectious KoRV (or phaCin-beta) vs. effects of the proviral insertion on the host gene. The effects of a given ERV may reflect either or both of these processes. The ERVs associated with altered cancer incidence in Table 1 and those associated with reproductive success in Table 2 are interesting. The written narrative (351-413) is impeded by description of the functions of many of the host genes at the insertion sites. It would be clearer to focus on a few ERVs where there is a strong case for biological importance, e.g. the insertions in SLC29A1 and BCL2L1. The other host sites could be grouped into proto-oncogenes, tumor suppressor genes, etc. and simply listed by name. For instance the description of the function of LZTS1 on lines 358-360 could be eliminated.

Response: **We thank the reviewer for this suggestion. We changed results accordingly.**

4. Finally, it would be helpful for the authors to set their studies into a larger biological/evolutionary framework in the Discussion. On the one hand their results demonstrate significant effects of ERVs (particularly KoRVs) on cancer and reproductive capacity on koalas, as well as high ongoing endogenization of new ERVs. On the other hand, they now calculate that KoRV (the most recent koala ERV) may have begun to infect at least some koalas as far back as 300,000 years ago, concomitant with emergence of modern koalas. And KoRV as an infectious agent may be spreading from north to south in Australia. What are the implications for the wild koala population?

Response: **We changed discussion accordingly adding a paragraph on implication for wild population: “Considering that the process of endogenization has been occurring in koalas for at least the last 300,000 years, and that this is a natural event responsible for shaping 5 to 10% of mammalian genomes, it does not immediately warrant intervention in the wild. However, when other threats—such as climate change and habitat loss caused by human activity—are added to the equation, the situation worsens. Therefore,**

monitoring is recommended to allow genetic drift and natural selection to proceed. Following this, further evaluation of whether intervention is possible or practical should be conducted, especially for critically endangered populations. Nonetheless, for captive populations that serve as gene reserves for the species, it becomes crucial to prevent an increase in the frequency of deleterious ERVs.”

Reviewer #3 (Remarks to the Author):

Neumann and team have presented a manuscript that significantly advances our understanding of key retroviruses in the koala. Currently the koala is unique in that it is experiencing germline invasion by two viruses, KoRV and phaCin-B. The study of the process of invasion and inheritance by these viruses provides key understandings to not only the koala but much more broadly. The authors have significantly advanced previously published work by using a powerful set of 111 koala samples from family trees of koalas with up to three known generations. This enabled them to follow inheritance of key insertion sites in the koala genome. The use of these multi-generational pedigrees combined with key health data on both living and deceased koalas enabled the authors to link key insertional sites with health predictors. Finally, the key retroviral insertions could be associated with health signals such as leukemia, fertility and longevity, enabling the authors to develop a genetic risk score, which will be very useful for breeding of koalas in captivity in particular.

The authors have presented a significant amount of raw data, both in the main manuscript and also in Supplementary files, and this data strongly supports their study and their conclusions. The science is high quality and the methodology is strong and very well structured.

This study is the first comprehensive analysis of such a large and well structured pedigree to further characterise koala retrovirus integrations and their positive and negative selections.

One minor error on Line 445. The Lone Pine Koala Sanctuary is located in Brisbane, not on the Gold Coast of Queensland Australia.

Response: Thank you for the comment. We corrected that to Brisbane.

I strongly support the publication of this manuscript.

We thank the reviewer for taking the time to review our manuscript and for their support for publication.

Reviewer #3 (Remarks on code availability):

This is outside my area of expertise

RESPONSE TO REVIEWERS' COMMENTS

Reviewer #1 (Remarks to the Author):

I am happy to support publication of this manuscript in revised form although I still feel that a degree of simplification by focussing on the KoRVs would be advantageous. However, I do fully understand the authors' wish to include all the available data in one place.

I have a few minor comments/corrections:

L84. Insert 'potentially' before deleterious

L94. More than?

L306. 'Expected' or 'accepted' rather than excepted

L529. Where does the figure of 300,000 come from? The phaCin- β (like) elements are clearly older while the KoRV data seems rather soft ("younger estimated dates cannot be discarded"-L166)

We thank the reviewer for their kind revision and appreciate their understanding in our decision to keep all the results together. All suggested modifications were applied to the final version of the manuscript.

The figure of 300,000 years comes from our current estimation of 312,191 years for KoRV. That is true, our conclusion was miswritten when considering that endogenization of phaCin- β (like) elements is much older. We changed it now to 6.9 Mya. And we do agree that the estimation to KoRV colonization is still soft and is open for discussion. But most importantly is that it is a recent event in comparison to the other retroviruses.

Reviewer #2 (Remarks to the Author):

In their revised manuscript, the authors have addressed the questions raised in my previous review and modified the presentations where appropriate. I recommend publication of this work. It is comprehensive, novel and appropriate for this journal.

Thank you.